# Comparative Analysis of Biochemical Parameters, Thermal Behavior, Rheological Features, and Gelling Characteristics of Thai Ligor Hybrid Chicken and Broiler Meats

**DOI:** 10.3390/foods14010055

**Published:** 2024-12-27

**Authors:** Ngassa Julius Mussa, Chantira Wongnen, Warangkana Kitpipit, Worawan Panpipat, Mingyu Yin, Siriporn Riebroy Kim, Manat Chaijan

**Affiliations:** 1Food Technology and Innovation Research Center of Excellence, School of Agricultural Technology and Food Industry, Walailak University, Nakhon Si Thammarat 80160, Thailand; ngassajulius@gmail.com (N.J.M.); chantira.wo@wu.ac.th (C.W.); warangkana.ki@wu.ac.th (W.K.); pworawan@wu.ac.th (W.P.); 2Akkhraratchakumari Veterinary College, Walailak University, Nakhon Si Thammarat 80160, Thailand; 3College of Food Science and Technology, Shanghai Ocean University, No.999, Huchenghuan Rd., Pudong New District, Shanghai 201306, China; myyin@shou.edu.cn; 4Food and Nutrition Program, Faculty of Agriculture, Kasetsart University, Bangkok 10900, Thailand; siriporn.r@ku.th

**Keywords:** hybrid, chicken, gel, quality, meat

## Abstract

Genetic differences typically cause differences in the structure and function of proteins in meat. The objective of this research was to examine the biochemical characteristics and functional behavior of proteins in fresh composite meat from Thai Ligor hybrid chicken (LC) and commercial broiler chicken (BC). The composite meat samples, which comprise minced breast and thigh without skin from 20 chicken carcasses in a 1:1 (*w*/*w*) ratio, were randomly selected for analysis using the completely randomized design (CRD). Results showed that BC meat exhibited higher ultimate pH after 24 h, Ca^2+^-ATPase activity, and trichloroacetic acid (TCA)-soluble peptide content compared to LC meat (*p* < 0.05). While both meat types showed non-significant differences in reactive sulfhydryl (SH) levels (*p* > 0.05), LC meat exhibited higher hydrophobicity compared to BC meat (*p* < 0.05). Differential scanning calorimetry (DSC) analysis revealed a single transition peak in all samples. LC meat exhibited higher thermal stability than BC meat, with transition peaks at 91 °C and 81 °C, respectively, in non-sodium chloride (NaCl) treated samples. Samples treated with 2.5% NaCl exhibited transition peaks around 70 °C for BC and 79 °C for LC. LC meat showed higher storage modulus (G′) and loss modulus (G″) values than BC meat, suggesting a stronger gel-forming tendency. LC meat gels exhibited higher hardness, cohesiveness, gumminess, and chewiness, and a slightly lower pH (6.14 vs. 5.97) compared to BC meat gels (*p* < 0.05). LC meat gels displayed larger expressible moisture content (*p* < 0.05), although the value was approximately 6%. Compared to LC meat gels, BC meat gels appeared slightly whiter (*p* < 0.05). To compare the lipid oxidation of BC and LC meat gels day by day, the thiobarbituric acid reactive substances (TBARS) of gels stored at 4 °C in polyethylene bags were measured on Days 0, 4, and 8. Both BC and LC meat gels showed acceptable lipid oxidation-based rancid off-flavor after short-term storage at 4 °C, with TBARS values below 2 mg malondialdehyde (MDA) equivalent/kg on Day 8. Understanding these variations in biochemical properties and functional behavior can help optimize processing methods and produce meat products of superior quality that meet consumer preferences.

## 1. Introduction

Modern consumers are increasingly health-conscious, seeking foods with high nutritional value and lower fat and cholesterol content [1]. Chicken meat, an affordable and versatile protein source, aligns well with these preferences [2,3]. Additionally, its suitability for various culinary preparations and absence of religious restrictions contribute to its widespread acceptance [2,3]. Broiler chicken (BC) meat is a widely consumed commercial poultry product. Modern breeding techniques have enabled rapid growth, with broilers reaching market weight in as little as 5–6 weeks [4]. However, their higher fat content, particularly in the breast and thigh, compared to indigenous chickens [4], may deter health-conscious consumers. Indigenous chickens, on the other hand, are prized for their unique flavor, chewy texture, and high nutritional value. These slow-growing birds, often considered a delicacy in Asian countries, take 14–23 weeks to reach market weight [5]. Despite growing consumer demand, their production is limited by factors such as non-uniform size, slow growth rate, and irregular quality, leading to higher production costs [6].

To address these challenges, various countries, including China, India, Pakistan, Bangladesh, Korea, and Thailand, have developed crossbreed chickens that combine the desirable traits of both BC and indigenous breeds [3,7]. The Ligor hybrid chicken (LC), a promising new breed developed in Thailand, exemplifies this innovation by merging the flavor and nutritional value of Dang Suratthani indigenous chickens with the faster growth rate of SUT 101 (a crossbreed between broiler and layer chickens) dams [3]. This unique combination results in a chicken that offers improved growth performance, leaner meat with higher protein content, and a distinct amino acid profile. While some textural and color differences may exist, LC meat presents a nutritious and flavorful alternative to traditional BC [3].

The increasing demand for ready-to-eat meat products, driven by changing lifestyles and urbanization, has led to a surge in chicken meat consumption [8]. Chicken’s neutral flavor, consistent texture, and light color make it an ideal candidate for various processed products such as burgers, sausages, and nuggets. These products rely heavily on the functional properties of meat proteins, including solubility, water retention, emulsification, and gelation [9,10]. While in vitro studies have provided foundational knowledge on these properties, further research is required to assess these functionalities in different chicken breeds and optimize processing techniques for higher-quality products [11,12,13,14].

Notably, previous research highlights significant differences in the quality characteristics between native chicken and commercial BC, which may influence protein functionality, such as gelation—a key factor in the texture and stability of processed meat products [3]. However, a detailed understanding of these differences remains limited. The hybrid nature of LC introduces genetic and biochemical diversity that could affect the functionality of its proteins compared to BC. Investigating these properties is essential to uncover their impact on meat quality, enabling tailored processing approaches that meet consumer demands for taste, nutrition, and product stability.

Consequently, this study aims to bridge the knowledge gap by investigating the biochemical properties and functional behavior of proteins in composite meat from LC and BC. It hypothesizes that genetic variances between LC and BC result in distinct biochemical properties and protein functionalities. These insights are crucial for optimizing processing techniques and developing innovative meat products that align with modern consumer preferences for healthier, high-quality options.

## 2. Materials and Methods

### 2.1. Chemical

Chemicals and reagents, such as adenosine 5′-triphosphate (ATP), trichloroacetic acid (TCA), calcium chloride (CaCl_2_), sodium chloride (NaCl), potassium chloride (KCl), thiobarbituric acid (TBA), 5,5′-dithiobis (2-nitro-benzoic acid) (DTNB), and 1,1,3,3-tetramethoxypropane (TEP), were sourced from Sigma-Aldrich (St. Louis, MO, USA).

### 2.2. Chicken Meat

Twenty 12-week-old LC carcasses with an average live weight of 1.6 ± 0.1 kg (Figure 1a) and twenty 6-week-old BC carcasses with similar live weights (Figure 1b) were purchased from Smart Farm at Walailak University, Thailand. The chickens were slaughtered in accordance with traditional Halal practices. The carcasses were subsequently packed in a plastic box filled with ice (0–2 °C) and brought to the laboratory within an hour. Upon arrival, the breast and thigh without skin were taken from the carcasses and combined at a 1:1 (*w*/*w*) ratio to form a composite meat. The composite meat was chopped with a bowl cutter (Talsa K15e, The Food Machinery Co., Ltd., Kent, UK) to form a homogeneous composite sample with coarse particle consistency. The fresh composite samples were then analyzed for their biochemical properties and gelation functionality. This investigation used three different composite samples from separate batches.

### 2.3. pH Determination

The composite chicken meat samples were homogenized with distilled water at a 1:5 weight-to-volume ratio. The pH of the homogenates was then measured with a pH meter (EUTECH PH700, M/s Eutech Instruments, Singapore) [3].

### 2.4. Ca^2+^-ATPase Activity

The Ca^2+^-ATPase activity of natural actomyosin (NAM) from composite meat was evaluated using the method described by Panpipat and Chaijan [15]. The NAM was prepared according to the process outlined previously, with 0.6 M KCl as the solubilizing agent [16]. The produced NAM was diluted to 2.5–8 mg/mL with 0.6 M KCl (pH 7.0). One milliliter of NAM solution was mixed with 0.6 mL of 0.5 M Tris-maleate (pH 7.0) and 1 mL of 0.1 M CaCl_2_. Deionized water was added to bring the total volume to 9.5 mL. Thereafter, 0.5 mL of 20 mM ATP was introduced. The reaction was halted by adding 5 mL of cold 15% (*w*/*v*) TCA after 8 min of incubation at 25 °C. Following that, an RC-5B plus centrifuge (3500× *g*/25 °C/5 min) was used, and the content of inorganic phosphate (Pi) released in the supernatant was measured [17]. The Ca^2+^-ATPase activity was measured in µmoles Pi released/mg protein/min. Prior to the addition of ATP, cold TCA was used to prepare a blank solution.

### 2.5. Reactive Sulfhydryl (SH) Content

Reactive SH content was determined using the DTNB method [18]. The NAM sample (0.5 mL, 4 mg/mL) was mixed with 4.5 mL of 0.2 M Tris-HCl buffer at pH 6.8. After that, 0.5 mL of 0.1% DTNB solution was added to the mixture and incubated at 40 °C for 25 min. The absorbance was measured at 412 nm using a Shimadzu UV-2100 spectrophotometer (Shimadzu Scientific Instruments Inc., Columbia, MD, USA). A blank was created by replacing the sample with 0.6 M KCl at pH 7.0. The SH concentration was calculated using the molar extinction of 13,600 M^−1^ cm^−1^ and reported as mol/10^8^ g protein.

### 2.6. Hydrophobicity

The hydrophobicity of NAM in 20 mM phosphate buffer (pH 6) was assessed using bromophenol blue sodium salt (BPB) [19]. A mixture of 1 mL NAM suspension and 200 µL of BPB solution (1 mg/mL in distilled water) was prepared. A control sample, without NAM, was included for comparison. The mixtures were agitated at room temperature (26–28 °C) for 10 min. Following centrifugation at 2000× *g* for 15 min at room temperature, the absorbance (A) of the supernatant was recorded at 595 nm using a Shimadzu UV-2100 spectrophotometer. Phosphate buffer served as the blank. The amount of BPB bound was calculated using the following formula:(1)$$BPB\;bound\left(µg\right)=200\;µg\times \left[\frac{A_{control}\;-\;A_{sample}}{A_{control}}\right].$$

### 2.7. TCA-Soluble Peptide

A 2 g portion of the finely chopped sample was combined with 18 mL of 5% (*w*/*v*) TCA and homogenized for 2 min at 11,000 rpm using an IKA^®^ T25 digital Ultra-Turrax^®^ homogenizer (Staufen, Germany). The mixture was then incubated at 4 °C for 1 h before being centrifuged at 8000× *g* for 5 min using a Sorvall RC5B Plus centrifuge (Norwalk, CT, USA) [20]. The TCA-soluble peptides in the supernatant were quantified using the Lowry method [21] and expressed as µmol tyrosine per gram of sample.

### 2.8. Thermal Property

The thermal characteristics of raw meat samples were evaluated using differential scanning calorimetry (DSC) using a Perkin Elmer Differential Scanning Calorimeter [22]. The temperature was calibrated using an indium thermogram. The meat samples (12–20 mg) with and without 2.5% (*w*/*w*) NaCl were carefully weighed, deposited in aluminum pans, and sealed securely. The scanning was carried out at a heating rate of 2 °C/min across a temperature range of 20–100 °C. Prior to scanning, the system was equilibrated at 20 °C for 5 min with ice water as the cooling agent. An empty aluminum pan was utilized as a reference. The transition temperature was determined by recording the temperature at each peak maximum.

### 2.9. Rheological Property

The rheological parameters of meat samples with and without 2.5% (*w*/*w*) NaCl were determined with a HAAKE MARS 60 Rheometer (Thermo Fisher Scientific Inc., Yokohama, Japan), according to Somjid et al. [19]. Approximately 0.5 g of the sample was evenly distributed onto the sample holder using a parallel plate geometry with a diameter of 35 mm. The space between the plate and the holder was maintained at 0.5 mm, and a thin coating of oil was applied to avoid dehydration. The heating process was conducted at a constant frequency of 1 Hz and a strain amplitude of 2%, ensuring the system remained within the linear viscoelastic range. The temperature was increased from 20 to 90 °C at a rate of 2 °C/min, while changes in rheological properties, including the elastic or storage modulus (G′) and viscous or loss modulus (G″), were measured.

### 2.10. Gelation Functionality and Analyses

A gel was prepared using the procedure described by Somjid et al. [19]. To generate a homogenous sol, samples with approximately 80% moisture content were sliced into small portions, mixed with 2.5% (*w*/*w*) NaCl as a myofibrillar protein-solubilizing agent, and chopped for 5 min. Then, the sol was put inside a 2.5 cm-diameter polyvinylidene casing, which was securely tied on both ends. Following 30 min of incubation at 40 °C, the sol was cooked for 20 min at 90 °C. Before being analyzed, the gels were promptly cooled in ice water for 30 min after heating and then kept at 4 °C for 24 h (overnight).

#### 2.10.1. Textural Profile Analyses (TPA)

The TPA of the gels was performed following the method described by Li et al. [23], using a TA-XT2i texture analyzer (Stable Micro Systems Ltd., Godalming, UK). The texture analyzer was equipped with a cylindrical probe (P/50, 50 mm stainless steel cylinder) and a 25 kg load cell. Gel samples were prepared in cylindrical shapes with dimensions of 2.0 cm in height and 2.5 cm in diameter. A double-compression cycle test was conducted, compressing the samples to 50% of their initial height, with a 1 s interval between compressions. The trigger force for the test was set at 5 g, and the compression speed was maintained at 5 mm/s. The TPA parameters, including hardness (maximum compression force), springiness (the sample’s ability to regain its original shape after deformation), cohesiveness (the extent of deformation the sample withstands before breaking), gumminess (the energy required to ingest semisolid food), and chewiness (the work required to prepare a solid sample for swallowing), were calculated using the TA-XT Express software (version 19.0).

#### 2.10.2. Expressible Drip

In short, a 0.5 cm thick gel sample was weighed and positioned between 2 layers of Whatman filter paper No. 1, with 2 layers on top and three layers underneath. A standard weight of 5 kg was applied to the sample for 2 min. Afterward, the sample was taken out, and its weight was measured again. The expressible drip was calculated as a percentage of the initial sample weight [19].

#### 2.10.3. pH

The previously described method (Section 2.3) was used to measure the pH of the gels.

#### 2.10.4. Whiteness

Colorimetric values of gels, including *L** (lightness), *a** (redness/greenness), and *b** (yellowness/blueness), were examined with a portable Hunterlab Miniscan/EX instrument (10° standard observers, illuminant D65; Hunter Assoc. Laboratory, Reston, VA, USA). The instrument was standardized using calibrated black and white standards. The sample was placed in a glass sample cup and covered with an opaque cover to prevent ambient light from reaching the detector before the *L**, *a**, and *b** readings were taken. The gel whiteness was then estimated using the following formula [19]:(2)$$Whiteness=100\;-\;\sqrt{{\left(100-L*\right)}^{2\;}+{a*}^{2}+{b*}^{2}}\;.$$

#### 2.10.5. Thiobarbituric Acid Reactive Substances (TBARS)

Using the TBARS assay, which follows the procedure outlined by Buege and Aust [24], the production of secondary lipid oxidation products of gels stored at 4 °C in polyethylene bags was assessed at Days 0, 4, and 8 in order to compare the lipid oxidation of BC and LC meat gels day by day. In short, 2.5 mL of a TBARS solution containing 0.375% TBA, 15% TCA, and 0.25 N HCl was combined with the sample (0.5 g). The mixture was centrifuged at 5000× *g* for 10 min at 25 °C after being boiled in boiling water for 10 min to turn pink and then cooled with running tap water. The supernatant’s absorbance was measured at 532 nm. Using TEP (0 to 10 ppm), a standard curve was created, and TBARS was reported in milligrams of malondialdehyde (MDA) equivalents/kg.

### 2.11. Statistical Analysis

This study employed a Completely Randomized Design (CRD). For statistical analysis, SPSS for Windows (SPSS Inc., Chicago, IL, USA) was used. A paired *t*-test was performed to compare the means of all analyses conducted on meat samples from the LC and BC. In the case of TBARS, the values of LC and BC meat gels were also compared on a daily basis. Data were considered statistically significant if *p* < 0.05. Three replications of each parameter’s data were collected. The mean plus standard deviation (SD) is used to display the results.

## 3. Results and Discussion

### 3.1. pH

The ultimate pH of the meat is an essential contributor to the technological quality attributes of broiler meat and is profoundly affected by genetics [25,26,27]. Table 1 shows that BC meat had an ultimate pH of 6.09 after 24 h, while LC meat had a pH of 5.91 (*p* < 0.05).

Previous research also found that BC meat had a higher pH than meat from Thai indigenous chicken [28] and hybrid chicken [4]. It was discovered that the pH of Korat hybrid chicken breast and thigh meats reduced as the animal grew older [4]. In the present study, slow-growing LC (aged 12 weeks) were older than fast-growing BC (aged 6 weeks) [3]. According to Díaz et al. [29], the meat from older animals exhibited a lower pH compared to that of younger animals, which was attributed to an increased concentration of muscle glycogen. The glycogen levels in muscle are largely determined by shifts in the composition of muscle fibers, as their metabolic activities differ. For instance, in older birds, the breast and thigh muscles generally contained greater glycogen reserves, leading to a decrease in postmortem pH.

According to Zhang and Barbut [30], normal breast meat has an average ultimate pH of 5.7 to 5.9, and values greater than 6.1 or less than 5.7 are related to the development of meat quality defects such as DFD and acid meat. Herein, the mixed meat homogenate was examined in relation to the gel product formed from whole chicken meat. In the current study, the pH of BC meat was slightly higher than that of LC (+0.18 pH units), and the pH values, particularly from LC, were within the previously reported normal range of 5.7 to 5.9 [25,31,32].

The results were in agreement with those of Wei et al. [33], who found that Taihe silky chicken (TSC; a Chinese native chicken) meat had a lower ultimate pH than Cobb chicken meat. A lower pH in TSC meat was significant since it inhibited bacterial growth, resulting in a longer shelf life; however, it also meant that the meat had a lower water holding capacity (WHC) [33]. Another study found that the meat of slow-growing local breed chickens had a lower pH than fast-growing commercial BC, which could be attributed to fast-growing chickens’ reduced muscle glycogenolysis potential, primarily due to differences in glycogen content in the muscle [34].

### 3.2. Ca^2+^-ATPase Activity, Reactive SH Content, Hydrophobicity, and TCA-Soluble Peptide

Myofibrillar protein makes up more than half of meat proteins and is critical in the creation of desirable meat or meat products [35,36]. Myofibrillar protein has been shown to impact meat ultrastructure, which is often associated with meat product color, texture, and WHC [37]. As a result, the relevant indices to characterize the myofibrillar protein structure of chicken meat were determined herein, namely Ca^2+^-ATPase activity, an indicator to assess myosin integrity in myofibrillar protein [38], reactive SH content, and surface hydrophobicity, the parameters related to the protein unfolding upon gel-forming of myofibrillar protein [35]. Once myosin unfolded, the internal hydrophobic and SH groups were exposed, causing the protein to denature and form gels [35]. Proteolysis activity caused by endogenous proteases can be measured using TCA-soluble peptide content. The more the proteolysis activity, the lower the gel-forming ability of muscle foods can be predicted due to the difficulty in cross-linking to form a three-dimensional network of the proteins [39]. Many endogenous proteases are recognized as important in muscle proteins during storage and processing [40].

Ca^2+^-ATPase activity is a biological indicator of myosin molecules’ stability and integrity, and it is linked to contractile systems [18]. Loss of myosin integrity caused by denaturation and/or aggregation due to structural changes in myofibrillar proteins was connected to a decrease in Ca^2+^-ATPase activity [19]. In this study, BC meat exhibited significantly higher Ca^2+^-ATPase activity (0.51 µmol/mg protein/min) than LC meat (0.29 µmol/mg protein/min) (*p* < 0.05; Table 1). Ca^2+^-ATPase activity differences were caused by genetic variance between two breeds, as well as myosin’s susceptibility to denaturation or structural changes during muscle conversion to meat during slaughtering, cutting, and mincing.

SH groups are the most reactive functional groups found on the surface of protein molecules, and they influence protein functionalities such as gelation [38,41]. Furthermore, they have a significant impact on the WHC and texture of a gel, increasing proportionally with increases in hardness and WHC [18,42,43]. Furthermore, SH groups in the head section of myosin have an important role in Ca^2+^-ATPase function, particularly in strengthening the gel [18]. In the present study, both BC and LC meats showed non-significant reactive SH levels (2.66–2.87 mol/10^8^ g protein) (*p* > 0.05; Table 1). This SH content was analyzed in the minced fresh meat. When the meat was subjected to heat-induced gelation, the changes in SH content in the heated sol may have differed or been the same, depending on the degree of protein denaturation and endogenous SH content that could be exposed to the surface. Then, it may affect the strength of the resulting gel. According to Li et al. [44] and Zhang et al. [35], the reactive SH groups of chicken meat protein increased dramatically from 40 °C to 60 °C, suggesting that actomyosin was unfolded and ready for inter-molecular interactions that could cause protein aggregation. The exposed SH groups become available for disulfide (S-S) formation and myosin aggregation [44,45,46]. Similarly, a decrease in SH groups in chicken, fish, and pork was linked to an increase in disulfide bonds [41,47,48]. Thus, the presence of SH groups in surimi and muscle proteins is regarded as important for gel strength, and a low quantity of SH groups leads to low gelling ability [49,50,51].

Hydrophobicity is one of the most useful aspects in estimating protein denaturation and structural change. The hydrophobic environment is altered as a result of the rupture of some non-covalent connections in protein molecules, as well as changes in native structure. In other words, surface hydrophobicity alterations imply conformational changes in protein structure. Unfolding protein molecules may expose hydrophobic groups from the protein’s core, contributing to an increase in hydrophobicity [41]. Hydrophobicity is the ability of nonpolar solutes to bond to one another in an aqueous environment [52]. The exposure of hydrophobic residues or un-folding (increase in hydrophobicity) during either thermal treatment of meat protein or other factors such as pH, time, and ionic strength is regarded as a prerequisite for protein aggregation formation as well as an indicator of protein denaturation and structural change as the hydrogen bonds with protein chains are disrupted [38,53,54,55,56]. In the current investigation, LC meat had higher hydrophobicity (121.21 µg BPB bound) than BC meat (82.85 µg BPB bound) (*p* < 0.05; Table 1). This hydrophobicity, like that of other reactive groups, can change during heat gelation as a result of unfolding and subsequent aggregation. Li et al. [44] found that heating chicken actomyosin at temperatures ranging from 20 °C to 60 °C greatly increased its surface hydrophobicity. The increase was connected with a change in actomyosin structure, which included exposing hydrophobic residues to the protein surface, leading to the creation of hydrophobic interactions that participated in the gelation process. Furthermore, Zheng et al. [57] proposed that the enhanced hydrophobicity of heated chicken breast myofibrillar proteins compared to raw chicken myofibrillar proteins was caused by denaturation and exposure of hydrophobic groups, which strengthened hydrophobic interactions. As a result, we hypothesize that larger hydrophobic contents in LC meat than BC meat reflect greater protein unfolding and denaturation. Not only that, but it also shows that when heated to gel, hydrophobic regions may interact more strongly, contributing to gel stability [46,58].

In terms of proteolysis as evaluated by TCA-soluble peptide, BC meat had a greater TCA-soluble peptide content than LC (7.71 vs. 5.03 µmol tyrosine/g) (*p* < 0.05; Table 1), implying that the higher peptide content in BC was owing to endogenous peptide content or autolysis postmortem. TCA-soluble peptides show the degree of proteolysis in meat, which can impact its gelling ability. Furthermore, endogenous proteases can cause the breakdown of proteins, resulting in gel softening [59]. It has been discovered that surimi gels weaken at temperatures ranging from 50 °C to 70 °C. This process is known as modori and is caused by endogenous heat-activated proteases that can hydrolyze myosin [60]. As a result, higher TCA-soluble peptide levels in BC meat indicate more proteases than LC meat. Furthermore, as previously observed, increased proteolysis of surimi after heat treatment weakens the gel, resulting in a decreased breaking force [61]. To reduce proteolysis activity, washing to remove some proteases as well as soluble oligopeptides, such as when producing surimi or other surimi-like materials, applying protease inhibitor additives, and heating at the appropriate temperature are viable options [59,60,61].

According to the findings, LC meat may form gel more easily during thermal gelation due to lowered myosin integrity, making it possible for simpler unfolding, as well as a large increase in hydrophobicity, which can enable hydrophobic interaction within the gel-forming network. Both BC and LC contained the same reactive SH content, which could oxidize to create disulfide bonds to the same extent during thermal gelation. Furthermore, LC had a reduced TCA-soluble peptide content, which suggested fewer potential protein fragments, allowing the three-dimensional network to assemble more strongly during gelation.

### 3.3. Thermal Property

Temperature is the most significant element influencing the gelling characteristics of myofibrillar proteins since heat is needed for denaturation, unfolding, and irreversible protein aggregation in order to create a gel. DSC has been widely utilized to investigate the heat stability and behavior of protein structures in muscle foods and processed products [62]. The DSC thermograms of chicken meat samples treated with or without 2.5% (*w*/*w*) NaCl of LC and BC were measured (Figure 2). Because of its multifunctionality, NaCl is commonly employed in the development of muscle-based gel products to improve muscle protein solubility and therefore gelation, as well as to enhance the flavor and shelf life of the products. However, reduced-salt or low-sodium products are becoming more popular among health-conscious consumers. So, the test was carried out to determine how NaCl affects the thermal behavior of chicken meat from both LC and BC, which may be used as a support strategy for the manufacturing of chicken meat-based gel products.

A single transition peak was observed in all samples, including chicken meat with and without NaCl (Figure 2), at varying transition temperatures. Differences in endothermic transitions were due to different sample scenarios [43]. In Figure 2a of the non-NaCl-treated sample, BC had the transition peak at 81 °C and LC had the peak at 91 °C, which was associated with the denaturation of their muscle proteins. In chicken and other meat, substantial peak transitions are seen in the DSC thermogram, indicating major protein denaturation such as myosin, actin, sarcoplasmic proteins, and collagen, with varied patterns [43]. The results indicated that LC meat was more thermostable than BC meat. This resistance may be connected to the genetic makeup of the avian myofibrillar protein [63]. It has been established that the thigh, breast, and skin of LC have more collagen content than BC [3], which may be related to the higher transition temperature of LC.

As shown in Figure 2b, samples treated with 2.5% NaCl exhibited transition peaks at around 70 °C for BC and 79 °C for LC. Notably, adding NaCl resulted in lower transition temperatures for both meats. This was attributed to the denaturing impact of salt, which can aid in the thermal denaturation of muscle proteins.

It has been reported that NaCl treatment (0.70%) significantly reduced the denaturation temperature (Td) of myosin and actin in both whole muscle and isolated myofibrillar proteins from bovine meat, indicating that NaCl destabilized myosin and actin, making the protein more prone to thermal denaturation (Td decreases) [64]. It was postulated that the incorporation of salt causes anions to compete with water molecules for certain protein locations, affecting their hydration characteristics and necessitating lower denaturation energies [64]. The pattern of the BC DSC thermogram was significantly altered after the addition of NaCl. After the initial peak was recorded, the heat flow increased consistently throughout the heating regime. This could be related to the aggregation of BC muscle components once denatured at 70 °C.

### 3.4. Dynamic Rheological Properties

Dynamic rheology analysis is widely employed to investigate the viscoelastic properties and molecular interactions of muscle proteins during gelation, a critical process underlying texture development [19,65]. Figure 3 depicts the rheological properties of BC and LC pastes applied with and without 2.5% (*w*/*w*) NaCl after heating from 20 °C to 90 °C. G′ measures the strength (stored energy) when a gel is deformed, thus reflecting properties such as elasticity or solid-like, while G″ demonstrates the energy loss of a material due to viscous deformation, thus reflecting viscosity or liquid-like of a gel [66,67].

Changes in the G′ of samples without NaCl and with NaCl as a function of temperature are shown in Figure 3a,b, respectively. Different patterns of changes in G′ were found among samples, indicating the effect of both chicken breed and NaCl on the gel-forming ability. LC had a higher G′ than BC, especially in the sample without NaCl, suggesting the tendency of higher gel strength of LC. LC meat had more thermal stability, as found in the DSC result (Figure 2), thus the G′ remained stable or decreased a little bit during heating up to around 50 °C. Then, the G′ increased sharply and peaked at around 80 °C, before decreasing. This suggested that the thermal gelation of LC meat without NaCl occurred progressively during heating at 50–80 °C. The increase in G′ at the temperatures after 50 °C to 80 °C is an indication of the transition from a liquid-like state to a solid-like one or the formation of new bonds to produce a permanent network structure or irreversible myosin filaments or complexes [68]. When it was further heated up to 90 °C the G′ decreased, which may be due to the destabilization of some bonds, like hydrogen bonds, which are sensitive to high temperature, leading to the balance between protein-protein and protein-water interactions as found in the gel network in general. The G′ of BC meat without NaCl remained constant until roughly 42 °C, then steadily climbed to around 78 °C, with a minor drop thereafter. In this study, the lowest G′ can be initially seen at around 50 °C for LC and 42 °C for BC, both with and without NaCl, but it can clearly be seen in the sample without NaCl. A transient decrease in G′ has been noted in myofibrillar protein gels at temperatures ranging from 50 °C to 60 °C across various muscle sources, such as chicken and fish. This effect is attributed to the activity of residual endogenous proteases [69,70,71]. Since muscle tissue contains numerous endogenous proteases, those not eliminated during processing may contribute to gel weakening, partially explaining this behavior [39]. Cathepsin D, derived from sources such as bovine, chicken, ostrich, and pork, has been shown to remain active at elevated temperatures (33–60 °C) [72,73,74] and even exhibits substantial activity near 70 °C [75,76].

When NaCl was added to both samples, the G′ of both samples decreased, indicating that the gel’s texture became softer. NaCl can dissolve myofibrillar proteins, which are accountable for gel network formation. The G′ of both samples increased at 54 °C, reached its peak at 78–80 °C, and then stayed steady or declined somewhat. The staying or small decrease in G′ in the presence of NaCl heated at high temperature could be attributed to NaCl’s water-binding capacity, which may stabilize the gel structure. Increases in G′ indicate the development into an elastic gel network and reflect variations in the rigidity of the meat batter. G′ values can vary depending on species, muscle type, pH, and salt concentration [19,20,30].

For the G″ (Figure 3c,d), the values from both samples without NaCl began to decrease from 20 °C to 46 °C (Figure 3c). This decrease is due to the domains within the α-helical portion of myosin of meat protein being involved in thermal unfolding and also suggests that the partial unfolding of myosin within that temperature range led to an increase in viscosity [77]. Subsequently, G″ rose, signifying the initiation of gelation or the development of an elastic protein network structure, linked to the unfolding and aggregation of myosin filaments [78]. The pattern of changes is similar to the G′ pattern; however, with the addition of NaCl (Figure 3d), G″ seemed to be lower than or equal to G′ in both samples, demonstrating the production of elastic gel. A typical gel is formed when G′ is much larger than G″ [79]. According to Xu et al. [80], greater G′ and G″ values indicate a stiffer or firmer gel network topology. As a result, with or without NaCl, LC meat is more likely to produce a firmer gel than BC meat.

When considering the effect of NaCl addition, it appeared that NaCl was still required to improve the gelation of both chicken meats because it can solubilize the key myofibrillar proteins, specifically myosin, which was important for gelation functionality. It has been established that the dissociation of myosin and actomyosin from myofibrils, as well as their form and structure, are critical to meat product quality. To achieve suitable WHC and binding capabilities after heat-induced gel formation, it is required to enhance the amount of eluted myosin and actomyosin while also controlling their extracted forms [81]. Thus, the gel-forming ability of both chicken meats was examined in the presence of 2.5% NaCl, as described in the following section.

### 3.5. Gelation Functionality

#### 3.5.1. TPA

Table 2 shows the textural characteristics of LC and BC meat gels as assessed by TPA. Hardness, springiness, cohesiveness, gumminess, and chewiness were all reported. LC meat gel had higher hardness, which is the force needed to compress a sample in order to achieve a specific deformation and is related to gel strength (*p* < 0.05). A similar pattern was observed for cohesiveness, which measures how well meat gel keeps together when compressed [82]. The gumminess and chewiness followed a similar pattern to the hardness. Gumminess is the result of hardness and cohesiveness, whereas chewiness includes springiness, gumminess, and the distance across which the food is deformed. Gumminess indicates the energy used to ingest semisolid food, whereas chewiness represents the energy needed to chew the food to the degree at which it can be swallowed [82].

Muscle fiber structure and connective tissue may influence the hardness and other textural aspects of chicken flesh gel, such as cohesiveness, gumminess, and chewiness. LCs are typically raised in more conventional, free-range environments that enable more physical exercise [3]. This may result in highly developed and thick muscle fibers, increasing the strength of the meat and its gel products. In contrast, BC are frequently raised in controlled conditions with little activity, resulting in more soft meat due to reduced muscle density and connective tissue development [3]. Furthermore, LC has been linked to slower growth rates, allowing for the formation of more collagen, which contributes to the meat’s hardness. Panpipat et al. [3] found that LC meat contained higher collagen content, especially in thigh muscle, compared to BC meat (*p* < 0.05). When compared to BC meat, the LC’s rough flesh and high collagen content are believed to be caused by the fact that the native Dang Surathani sires used to produce it were used for cockfighting [3,4,5]. Higher collagen concentration in LC meat may contribute to the gel’s harder texture. It has been reported that LC meat had more protein, moisture, and ash but less fat than BC meat (*p* < 0.05) [3]. The higher protein concentration of LC meat may also be linked to its gelation functionality. Springiness was not significantly different between the gels of the two species of chicken (*p* > 0.05). The capacity of the gel to regain its shape following distortion is referred to as springiness. Gel elasticity can be reflected in springiness; higher values indicate a homogenous and coordinated gel structure [83]. The similarity in this characteristic indicates that both species of chickens have comparable muscle elasticity, which could be attributable to a similar three-dimensional network as well as elastic behavior.

#### 3.5.2. pH, Expressible Moisture Content, and Whiteness

BC meat gel has a slightly higher pH than LC meat gel (pH_BC_ 6.14 vs. pH_LC_ 5.97; *p* < 0.05; Table 2), similar to the original pH of raw materials (pH_BC_ 6.09 vs. pH_LC_ 5.91; Table 1). A slight increase in the pH of gel when compared to raw meat was most likely caused by the release of soluble acidic compounds into the cooking water during thermal gelation, the formation of basic compounds in the gel during heating, and the loss of meat buffering capacity during processing.

Myofibrillar proteins have been reported to have a pI of around 5.0–5.5 [84]. The gel’s final pH may influence its strength and WHC. The LC gel had a pH of 5.97, which was closer to the pI than the BC gel. Thus, the LC meat may have more protein–protein interaction than protein-water interaction, resulting in more water release from the gel, as seen by a larger expressible moisture content (Table 2). However, the expressible drip content of LC meat was approximately 6%, indicating a gel with adequate WHC. Because BC meat usually has a higher pH, it may hold moisture more effectively in the gel matrix during gelation, which would reduce the amount of moisture that can be expressed. The gelation characteristics of chicken breast and thigh meat homogenates (4.5% protein) were found to be pH-dependent by Lesiow and Xiong [85]. The ideal gelling pH for breast meat homogenates (pH 6.30) was somewhat greater than that for thigh meat homogenates (pH 5.80–6.30) at the same protein concentration. In the present study, the whole muscle was used; therefore, the gelling behavior could differ from the specified meat portions, such as the breast and thigh.

Table 2 also displays the whiteness of the BC and LC meat gels. Compared to LC meat gel, BC meat gel was somewhat whiter (*p* < 0.05). One potential explanation was that whiteness is affected by a number of variables, including pigment concentration, muscle fiber type, and fat content. Fast-twitch (white) muscle fibers are lighter in color and tend to be more prevalent in BC. Because of their active lifestyle, LC may have more slow-twitch (red) fibers, giving them a significantly darker appearance [3]. Compared to BC, more red and yellow pigments were found in indigenous chicken muscles by Wattanachant et al. [28]. Furthermore, because fat reflects more light and makes things appear whiter, BC’s higher fat content may also be a factor in the whiter shade.

#### 3.5.3. TBARS

The initial TBARS readings of BC and LC meat gels were low (0.7–0.78 mg MDA equivalent/kg), and no significant differences were observed (*p* > 0.05), indicating the same oxidative status of lipids at the beginning. However, TBARS results for BC and LC meat gels increased over time, with BC having slightly higher values, particularly on Days 4 and 8. TBARS measures the degree of lipid oxidation, with higher levels indicating more oxidation. The somewhat higher TBARS levels in BC could be attributed to the higher fat content, which provides a larger substrate for oxidation [3]. In contrast, LC meat, which contains less fat, seems less susceptible to oxidation. Panpipat et al. [3] found that LC meat had greater moisture, protein, and ash content but lesser fat content than BC meat (*p* < 0.05). Furthermore, the slower growth and natural diet of LC may result in more natural antioxidants in the muscle, slowing the rate of lipid oxidation. The active lifestyle of LC may also promote healthier lipid profiles, which can oxidize more slowly if the meat contains enough antioxidants. TBARS values > 2 mg of MDA equivalent/kg indicate that the food is likely to be detected as rancid by consumers while also producing other unusual odors [86]. Domínguez et al. [87] recommended limiting meat and meat products to 2–2.5 mg MDA equivalent/kg to mitigate rancidity. The BC and LC meat gels had acceptable lipid oxidation-based rancid off-flavor after short-term storage at 4 °C, with TBARS values < 2 mg MDA equivalent/kg on Day 8.

## 4. Conclusions

The findings verified the hypothesis that genetic variations caused significant differences in the biochemical characteristics and functionality of proteins in composite meat between LC and BC. Understanding these biochemical and functional differences is crucial for optimizing processing techniques and developing high-quality products, particularly from LC meat, a newly developed breed. Future research should evaluate the sensory and nutritional profiles of LC meat gels to assess consumer suitability and explore their potential in novel products like emulsified meats and protein-rich snacks. These efforts will enhance the commercial value of LC meat and align with market demands.

## Figures and Tables

**Figure 1 foods-14-00055-f001:**
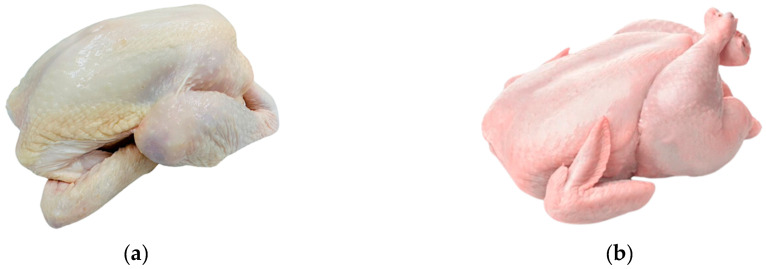
Carcasses of Ligor chicken (LC; (**a**)) and broiler chicken (BC; (**b**)).

**Figure 2 foods-14-00055-f002:**
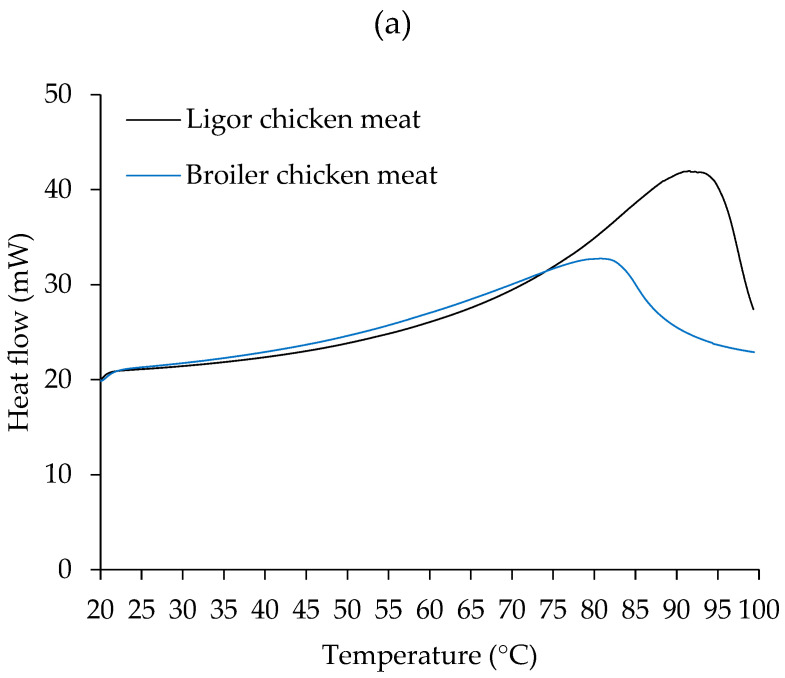
Differential scanning calorimetry (DSC) thermograms of Ligor chicken (LC) meat and broiler chicken (BC) meat without NaCl (**a**) and with 2.5% (*w*/*w*) NaCl (**b**).

**Figure 3 foods-14-00055-f003:**
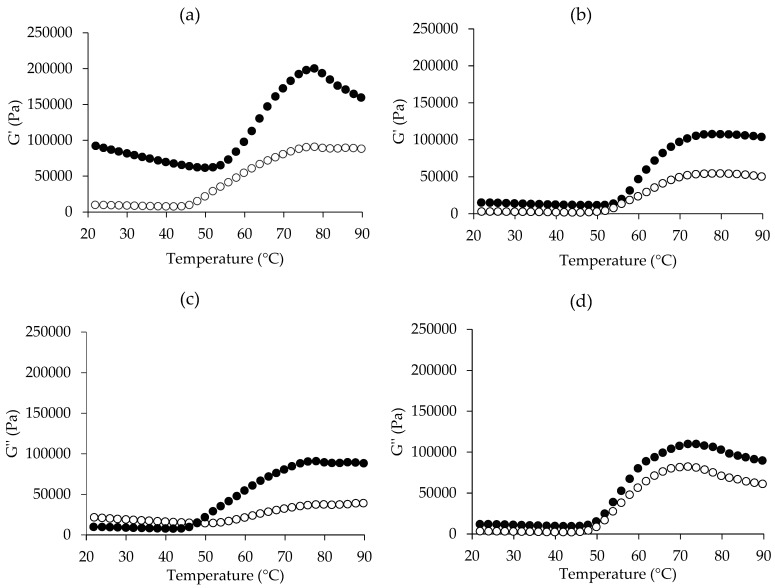
Rheological behaviors, including storage modulus (G′) and loss modulus (G″) of Ligor chicken meat paste (●) and broiler chicken meat paste (○) without NaCl (**a**,**c**) and with 2.5% (*w*/*w*) NaCl (**b**,**d**).

**Table 1 foods-14-00055-t001:** Muscle pH and some biochemical properties of Ligor chicken meat and broiler chicken meat.

Parameters	Ligor Chicken Meat	Broiler Chicken Meat	*p*-Value
pH	5.91 ± 0.09 ^b^	6.09 ± 0.04 ^a^	0.028
Ca^2+^-ATPase activity (µmol/mg protein/min)	0.29 ± 0.14 ^b^	0.51 ± 0.24 ^a^	0.022
Reactive sulfhydryl (SH) content (mol/10^8^ g protein)	2.66 ± 1.88 ^a^	2.87 ± 1.75 ^a^	0.227
Hydrophobicity (BPB bound; µg)	121.21 ± 1.56 ^a^	82.58 ± 6.27 ^b^	0.000
TCA-soluble peptide (µmol tyrosine/g)	5.03 ± 1.77 ^b^	7.71 ± 3.75 ^a^	0.001

Values are shown as mean ± standard deviation of triplicate measurements. Significant differences (*p* < 0.05) are considered between letters in the same row. BPB = bromophenol blue. TCA = trichloroacetic acid.

**Table 2 foods-14-00055-t002:** Appearance, texture profile analysis (TPA) parameters, expressible drip, pH, whiteness, and thiobarbituric acid reactive substances (TBARS) of Ligor chicken (LC) and broiler chicken (BC) meat gels.

Parameters	Ligor Chicken Meat	Broiler Chicken Meat	*p*-Value
Appearance	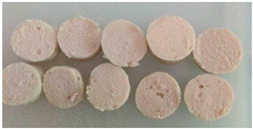	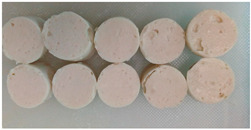	-
TPA *			
Hardness (N)	9.81 ± 1.16 ^a^	5.87 ± 0.38 ^b^	0.013
Springiness (cm)	0.11 ± 0.05 ^a^	0.10 ± 0.04 ^a^	0.887
Cohesiveness	0.39 ± 0.08 ^a^	0.35 ± 0.02 ^b^	0.016
Gumminess (N)	3.83 ± 1.27 ^a^	2.02 ± 0.19 ^b^	0.027
Chewiness (N.cm)	39.44 ± 16.64 ^a^	20.37 ± 2.75 ^b^	0.004
Expressible moisture content (%)	5.77 ± 2.64 ^a^	3.85 ± 0.12 ^b^	0.005
pH	5.97 ± 0.01 ^b^	6.14 ± 0.01 ^a^	0.002
Whiteness	55.33 ± 0.92 ^b^	57.57 ± 0.27 ^a^	0.008
TBARS (mg MDA equivalent/kg) **			
Day 0	0.75 ± 0.03 ^a^	0.78 ± 0.02 ^a^	0.096
Day 4	1.07 ± 0.05 ^b^	1.19 ± 0.02 ^a^	0.011
Day 8	1.15 ± 0.04 ^b^	1.23 ± 0.04 ^a^	0.043

Data are reported as mean ± standard deviation (SD) from triplicate measurements. Significant differences are shown by different superscripts within the same row (*p* < 0.05). * The gels’ texture profile analysis (TPA) was conducted using a texture analyzer equipped with a cylindrical probe (P/50, 50 mm stainless steel cylinder) and a 25 kg load cell. A double-compression cycle test was performed, compressing the samples to 50% of their original height. The interval between the two compression cycles was set to 1 s. The test was initiated with a trigger force of 5 g and conducted at a speed of 5 mm/s. ** TBARS were measured in gels kept at 4 °C in polyethylene bags on Days 0, 4, and 8. MDA = Malondialdehyde. Fresh raw material had similar TBARS values (0.05 ± 0.03 MDA equivalent/kg for LC meat and 0.05 ± 0.02 for BC meat).

## Data Availability

The original contributions presented in this study are included in the article. Further inquiries can be directed to the corresponding author.

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
