# Peer review of "Comparative Analysis of Biochemical Parameters, Thermal Behavior, Rheological Features, and Gelling Characteristics of Thai Ligor Hybrid Chicken and Broiler Meats"

_foods, 2024, doi:10.3390/foods14010055_

Round 1
Reviewer 1 Report
Comments and Suggestions for Authors
Abstract- The abstract must contain objective, brief introduction, brief material and methods, results and conclusion, in equally proportional quantities. The first three items are missing and must be included
Introduction - In the introduction, it is important that authors provide information about the parameters researched, which are described in the material and methods section. The introduction text provides a broader perspective on sensory analysis, but the methodology is more robust and in-depth. Therefore, it is recommended that authors include, in the introduction, the importance of deepening the study, linking it with the analyzes carried out
MeM - TBARS description needs to have a sub-heading
Results and discussion - It is important that the order in which the results are described is the same as that presented in the material and methods. Authors must write the results in the same order
In all tables and graphics, P-value must be added.
Conclusion - The conclusion must be completely re-written. In it, “the learning from the study” must be written, and there must be no repeat description of results
Author Response
Reviewer 1
Abstract- The abstract must contain objective, brief introduction, brief material and methods, results and conclusion, in equally proportional quantities. The first three items are missing and must be included
Ans: The abstract was updated accordingly. A brief introduction, materials and methods, key findings, and conclusions were all presented.
Introduction - In the introduction, it is important that authors provide information about the parameters researched, which are described in the material and methods section. The introduction text provides a broader perspective on sensory analysis, but the methodology is more robust and in-depth. Therefore, it is recommended that authors include, in the introduction, the importance of deepening the study, linking it with the analyzes carried out
Ans: The Introduction begins by emphasizing the significance of chicken meat in human nutrition, highlighting its affordability, versatility, and alignment with health-conscious consumer preferences. It then discusses broiler chickens (BC) as a widely consumed poultry product, outlining their advantages and limitations, particularly their higher fat content. This leads to the introduction of hybrid chickens, developed to combine the desirable traits of broilers and indigenous chickens. The newly developed Ligor chicken (LC), a Thai hybrid breed, is then highlighted for its unique combination of flavor and nutritional value. The discussion transitions to consumer demand for ready-to-eat meat products, driven by changing lifestyles, and links this demand to the functional properties of meat proteins, such as gelation, which are critical for processed products. The objective and hypothesis of the study are subsequently presented, proposing that LC and BC meats exhibit differences in biochemical properties and protein functionality due to genetic variances. The Introduction concludes by outlining the expected outcomes, focusing on the potential to optimize processing techniques and develop high-quality meat products that meet consumer expectations for taste, nutrition, and functionality.
MeM - TBARS description needs to have a sub-heading
Ans: Done.
Results and discussion - It is important that the order in which the results are described is the same as that presented in the material and methods. Authors must write the results in the same order
Ans: When we were preparing the manuscript, we were concerned about this. The results are therefore explained in the same sequence as the materials and methods.
In all tables and graphics, P-value must be added.
Ans: The p-values were already included into the Tables as suggested by Reviewer 2. We displayed the rheology graph and the DSC-thermogram for the Figures. Therefore, the p-value was left out.
Conclusion - The conclusion must be completely re-written. In it, “the learning from the study” must be written, and there must be no repeat description of results
Ans: The suggestions from both reviewers were considered, and the conclusion was updated accordingly.

Reviewer 2 Report
Comments and Suggestions for Authors
I believe that images would enrich the paper. As well as a graphical summary
Abstract:
Initial methodological details are important: what is the experimental design? How many samples were taken? How were the samples prepared? Was the analysis performed on the entire carcass or on one of the meat cuts?
Acronyms need to be defined before their use. Correct this in the abstract and throughout the text of the paper
Were some analyses performed over time, how was this information worked on? Was a factorial analysis performed? Detail this in the abstract and in the article's methodology
What is the research hypothesis? Include it in the introduction
Lines 87 - 88 - How were the birds slaughtered? Did any recommendations from the current agencies follow?
Lines 88 - 90 - What is the average live weight of the chickens? How were the birds slaughtered? Did any recommendations from the current agencies follow?
Lines 90-91 - What is the temperature at which the carcasses were stored? Line 93 - What is the particle size?
Lines 191-193 - What is the calibration equation? Was the device previously calibrated? How did it happen? Did the device operate in an open or closed cone? I suggest that the authors also present the saturation and hue
2.10.5 Thiobarbituric Acid Reactive Substances (TBARS) - I believe this information is not in the correct place
Line 195 - How was the statistical analysis of these analyses performed over time? Describe in the topic that the statistical analysis is detailed.
Lines 205-208 - Insert the statistical model used
Results and discussion
The probability values should be inserted in the tables
Table 2 - The methodology for the texture parameters should be inserted in detail
The conclusion presented is a summary of the article and this is not correct. The conclusion should respond to the research objectives and should be concise. In addition, it should be noted whether the research hypothesis was accepted and present a recommendation for future studies.
references must be checked
Author Response
Reviewer 2
I believe that images would enrich the paper. As well as a graphical summary
Ans: Carcasses of Ligor chicken (LC) and broiler chicken (BC) are given in Figure 1. A graphical abstract was also provided as an attachment in the submission system.
Abstract:
Initial methodological details are important: what is the experimental design? How many samples were taken? How were the samples prepared? Was the analysis performed on the entire carcass or on one of the meat cuts?
Ans: The details were given at the beginning of the Abstract. “The objective of this research was to examine the biochemical characteristics and functional behavior of proteins in fresh composite meat from Thai Ligor hybrid chicken (LC) and commercial broiler chicken (BC). The composite meat samples, which comprised of minced breast and thigh without skin from 20 chicken carcasses in a 1:1 (w/w) ratio, were randomly selected for analysis using the completely random design (CRD). Results showed that…..” as well as in the last section of the Abstract. “….To compare the lipid oxidation of BC and LC meat gels day-by-day, the thiobarbituric acid reactive substances (TBARS) of gels stored at 4 °C in polyethylene bags was measured on Days 0, 4, and 8. Both BC and LC meat gels showed……”
Acronyms need to be defined before their use. Correct this in the abstract and throughout the text of the paper
Ans: All of the acronyms in the abstract and throughout the text were defined when they were first used.
Were some analyses performed over time, how was this information worked on? Was a factorial analysis performed? Detail this in the abstract and in the article's methodology
Ans: All analyses were performed once in fresh samples, with the exception of lipid oxidation of the gel, which was performed at Days 0, 4, and 8 to compare two types of chicken meat gels, LC and BC, on a daily basis, as described in the Abstract and Methods.
What is the research hypothesis? Include it in the introduction
Ans: It was added in the last section of the Introduction “Consequently, this study aims to investigate the biochemical properties and functional behavior of proteins in composite meat from LC and BC, with the hypothesis that the biochemical properties and functionality of proteins from LC and BC meats differ due to genetic variances.”
Lines 87 - 88 - How were the birds slaughtered? Did any recommendations from the current agencies follow?
Ans: It was stated that “The chickens were slaughtered in accordance with traditional Halal practices”
Lines 88 - 90 - What is the average live weight of the chickens? How were the birds slaughtered? Did any recommendations from the current agencies follow?
Ans: The statement has been amended to reflect the relevant information. “Twenty 12-week-old LC carcasses with an average live weight of 1.6 ± 0.1 kg and twenty 6-week-old BC carcasses with similar live weights were purchased from Smart Farm at Walailak University in Nakhon Si Thammarat, Thailand. The chickens were slaughtered in accordance with traditional Halal practices. The carcasses were subsequently packed in a plastic box filled with ice (0-2 °C) and brought to the laboratory within an hour. Upon arrival, the breast and thigh without skin were taken from the carcasses and combined at a 1:1 (w/w) ratio to form a composite meat. The composite meat was chopped with a Talsa Bowl Cutter K15e (The Food Machinery Co., Ltd., Kent, UK) to form a homogeneous composite sample with coarse particle consistency. The fresh composite samples were then analyzed for their biochemical properties and gelation functionality. This investigation used three different composite samples from separate batches.”
Lines 90-91 - What is the temperature at which the carcasses were stored? Line 93 - What is the particle size?
Ans: The temperature was stated which was about 0-2 °C. The particle size was not measured, but we intended to create a homogeneous, coarse particle. Thus, it was stated: "The composite meat was chopped with a Talsa Bowl Cutter K15e (The Food Machinery Co., Ltd., Kent, UK) to form a homogeneous composite sample with coarse particle consistency."
Lines 191-193 - What is the calibration equation? Was the device previously calibrated? How did it happen? Did the device operate in an open or closed cone? I suggest that the authors also present the saturation and hue
Ans: Thank you very much. Whiteness is a measure of gel color from white muscle foods like fish and chicken. All three of the color parameters (L*, a*, and b*) in the equation were used to calculate this index. Therefore, for comparison, we chose whiteness to represent the color of the chicken gels.
The standardization and sample handling details were provided in the text. “The instrument was standardized using calibrated black and white standards. The sample was placed in a glass sample cup and covered with an opaque cover to prevent ambient light from reaching the detector before the L*, a*, and b* readings were taken”
2.10.5 Thiobarbituric Acid Reactive Substances (TBARS) - I believe this information is not in the correct place
Ans: It was moved to the correct place.
Line 195 - How was the statistical analysis of these analyses performed over time? Describe in the topic that the statistical analysis is detailed.
Ans: We stated in the section of TBARS determination that “Using the TBARS assay, which follows the procedure outlined by Buege and Aust [24], the production of secondary lipid oxidation products of gels stored at 4 °C in polyethylene bags was assessed at Days 0, 4, and 8, in order to compare the lipid oxidation of BC and LC meat gels day-by-day.” So, we compared the TBARS values of LC and BC on days 0, 4, and 8. However, to clarify, we have also detailed in the section on Statistical Analysis. “This study employed a Completely Randomized Design (CRD). For statistical analysis, SPSS for Windows (SPSS Inc., Chicago, IL, USA) was used. A paired T-test was performed to compare the means of all analyses conducted on meat samples from the LC and BC. In the case of TBARS, the values of LC and BC chicken meat gels were also compared on a daily basis. Data was considered statistically significant if p < 0.05. Three replications of each parameter's data were collected. The mean plus standard deviation (SD) is used to display the results.”
Lines 205-208 - Insert the statistical model used
Ans: It was stated that “This study employed a Completely Randomized Design (CRD).”
Results and discussion
The probability values should be inserted in the tables
Ans: The p-values were already included into the tables.
Table 2 - The methodology for the texture parameters should be inserted in detail
Ans: The methodology for analyzing TPA was included in the table's footnotes. “The gels' texture profile analysis (TPA) was conducted using a texture analyzer equipped with a cylindrical probe (P/50, 50-mm stainless steel cylinder) and a 25-kg load cell. A double-compression cycle test was performed, compressing the samples to 50% of their original height. The interval between the two compression cycles was set to 1 s. The test was initiated with a trigger force of 5 g and conducted at a speed of 5 mm/s.”
The conclusion presented is a summary of the article and this is not correct. The conclusion should respond to the research objectives and should be concise. In addition, it should be noted whether the research hypothesis was accepted and present a recommendation for future studies.
Ans: The conclusion was revised accordingly. Responses to the aim and hypothesis were provided. The major findings were highlighted, and recommendations for further investigations were offered.
“The findings verified the hypothesis that genetic variations caused significant differences in the biochemical characteristics and functionality of proteins in composite meat between LC and BC. BC meat exhibited a higher ultimate pH, Ca²⁺-ATPase activity, and water-holding capacity, as well as greater TCA-soluble peptide content. On the other hand, LC meat demonstrated higher surface hydrophobicity, superior thermostability, and better gel strength and texture properties. Both meat types showed comparable reactive SH levels and exhibited low lipid oxidation during short-term storage. Additionally, the inclusion of NaCl reduced the thermal transition temperatures of both LC and BC meats. Understanding these biochemical and functional differences is crucial for optimizing processing techniques and developing high-quality products, particularly from LC meat, a newly developed breed. Future research should evaluate the sensory and nutritional profiles of LC meat gels to assess consumer suitability and explore its potential in novel products like emulsified meats and protein-rich snacks. These efforts will enhance the commercial value of LC meat and align with market demands.”
references must be checked
Ans: The references were rechecked.

Round 2
Reviewer 1 Report
Comments and Suggestions for Authors
the introduction needs to provide more in-depth information about why the more in-depth analyzes were carried out. It is important to value the article, and for the reader to understand what they will get while reading. In the previous review, this was already requested.
the conclusion is the answer to the question asked in the objective. It needs to be described in a more direct and simplified way. As it stands, it looks like a new results and discussion text
Author Response
Reviewer 1
the introduction needs to provide more in-depth information about why the more in-depth analyzes were carried out. It is important to value the article, and for the reader to understand what they will get while reading. In the previous review, this was already requested.
Ans: The revised version better contextualizes the rationale by emphasizing the importance of the genetic and biochemical differences between LC and BC, and how these differences necessitate a deeper functional and biochemical analysis. It also explicitly links the research goals to the development of improved meat products.
the conclusion is the answer to the question asked in the objective. It needs to be described in a more direct and simplified way. As it stands, it looks like a new results and discussion text
Ans: The adjustment was performed as advised, by eliminating the results to minimize duplication with discussion.
“The findings verified the hypothesis that genetic variations caused significant differences in the biochemical characteristics and functionality of proteins in composite meat between LC and BC. Understanding these biochemical and functional differences is crucial for optimizing processing techniques and developing high-quality products, particularly from LC meat, a newly developed breed. Future research should evaluate the sensory and nutritional profiles of LC meat gels to assess consumer suitability and explore its potential in novel products like emulsified meats and protein-rich snacks. These efforts will enhance the commercial value of LC meat and align with market demands.”
